# COOPERATING RPN'S IMPROVE FEW-SHOT OBJECT DETECTION

## ABSTRACT

Learning to detect an object in an image from very few training examples - few-shot object detection - is challenging, because the classifier that sees proposal boxes has very little training data. A particularly challenging training regime occurs when there are one or two training examples. In this case, if the region proposal network (RPN) misses even one high intersection-over-union (IOU) training box, the classifier's model of how object appearance varies can be severely impacted. We use multiple distinct yet cooperating RPN's. Our RPN's are trained to be different, but not too different; doing so yields significant performance improvements over state of the art for COCO and PASCAL VOC in the very few-shot setting. This effect appears to be independent of the choice of classifier or dataset.

## 1 INTRODUCTION

Achieving accurate few-shot object detection is difficult, because one must rely on a classifier building a useful model of variation in appearance with very few examples. This paper identifies an important effect that causes existing detectors to have weaker than necessary performance in the few-shot regime. By remediating this difficulty, we obtain substantial improvements in performance with current architectures.

The effect is most easily explained by looking at the "script" that modern object detectors mostly follow. As one would expect, there are variations in detector structure, but these do not mitigate the effect. A modern object detector will first find promising image locations; these are usually, but not always, boxes. We describe the effect in terms of boxes reported by a region proposal network (RPN) (Ren et al., 2015), but expect that it applies to other representations of location, too. The detector then passes the promising locations through a classifier to determine what, if any, object is present. Finally, it performs various cleanup operations (non-maximum suppression, bounding box regression, etc.), aimed at avoiding multiple predictions in the same location and improving localization. The evaluation procedure for reported labeled boxes uses an intersection-over-union (IOU) test as part of determining whether a box is relevant.

A detector that is trained for few-shot detection is trained on two types of categories. Base categories have many training examples, and are used to train an RPN and the classifier. Novel categories have one (or three, or five, etc.) examples per category. The existing RPN is used to find boxes, and the classifier is fine-tuned to handle novel categories.

Now assume that the detector must learn to detect a category from a single example. The RPN is already trained on other examples. It produces a collection of relevant boxes, which are used to train the classifier. The only way that the classifier can build a model of the categories variation in appearance is by having multiple high IOU boxes reported by the RPN. In turn, this means that an RPN that behaves well on base categories may create serious problems for novel categories. Imagine that the RPN reports only a few of the available high IOU boxes in training data. For base categories, this is not a problem; many high IOU boxes will pass to the classifier because there is a lot of training data, and so the classifier will be able to build a model of the categories variation in appearance. This variation will be caused by effects like aspect, in-class variation, and the particular RPN's choice of boxes. But for novel categories, an RPN must report as many high IOU boxes as possible, because otherwise the classifier will have too weak a model of appearance variation – for example, it might think that the object must be centered in the box. This will significantly depress

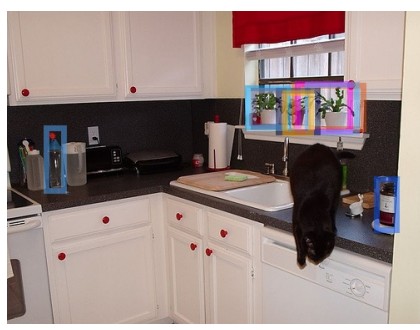 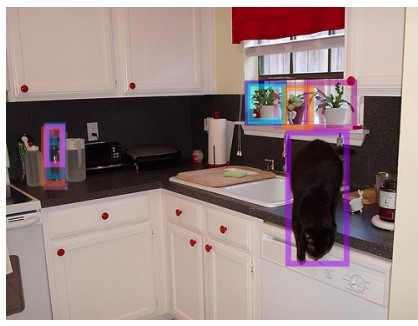

(a) TFA                                    (b) Ours

Figure 1: Few-shot detectors are subject to the *proposal neglect* effect. On the **left**: an image (from the PASCAL VOC split 3 novel classes test set) showing the top 10 proposals from a state-of-the-art few-shot detector TFA (Wang et al., 2020). On the **right**: the top 10 proposals from our cooperating RPN's. Cat is in the novel classes {boat, **cat**, mbike, sheep, sofa} (i.e., both models are **cat** detectors). Note that the classifier in TFA *will not see a cat box*, and so it cannot detect the cat. This is a disaster when this image is used in training. For our approach, the cat box is found and is high in the top 10 proposals.

accuracy. As Figure 1 and our results illustrate, this effect (which we call **proposal neglect**) is present in the state-of-the-art few-shot detectors.

One cannot escape this effect by simply reporting lots of boxes, because doing so will require the classifier to be very good at rejecting false positives. Instead, one wants the box proposal process to not miss high IOU boxes, without wild speculation. We offer a relatively straightforward strategy. We train multiple RPN's to be somewhat redundant (so that if one RPN misses a high IOU box, another will get it), without overpredicting. In what follows, we demonstrate how to do so and show how to balance redundancy against overprediction.

Our contributions are three-fold: (1) We identify an important effect in few-shot object detection that causes existing detectors to have weaker than necessary performance in the few-shot regime. (2) We propose to overcome the proposal neglect effect by utilizing RPN redundancy. (3) We design an RPN ensemble mechanism that trains multiple RPN's simultaneously while enforcing diversity and cooperation among RPN's. We achieve state-of-the-art performance on COCO and PASCAL VOC in very few-shot settings.

## 2 BACKGROUND

**Object Detection with Abundant Data.** The best-performing modern detectors are based on convolutional neural networks. There are two families of architecture, both relying on the remarkable fact that one can quite reliably tell whether an image region contains an object independent of category (Endres & Hoiem, 2010; van de Sande et al., 2011). In serial detection, a proposal process (RPN/s in what follows) offers the classifier a selection of locations likely to contain objects, and the classifier labels them, with the advantage that the classifier "knows" the likely support of the object fairly accurately. This family includes R-CNN and its variants (for R-CNN (Girshick et al., 2014); Fast R-CNN (Girshick, 2015); Faster R-CNN (Ren et al., 2015); Mask R-CNN (He et al., 2017); SPP-Net (He et al., 2015); FPN (Lin et al., 2017); and DCN (Dai et al., 2017)). In parallel, the proposal process and classification process are independent; these methods can be faster, but the classifier "knows" very little about the likely support of the object, which can affect accuracy. This family includes YOLO and its variants (for YOLO versions (Redmon et al., 2016; Redmon & Farhadi, 2017; 2018; Bochkovskiy et al., 2020); SSD (Liu et al., 2016); Cornernet (Law & Deng, 2018); and ExtremeNet (Zhou et al., 2019)). This paper identifies an issue with the proposal process that can impede strong performance when there is very little training data (the *few-shot* case). The effect is described in the context of serial detection, but likely occurs in parallel detection too.

**Few-Shot Object Detection.** Few-shot object detection involves detecting objects for which there are very few training examples. There is a rich few-shot classification literature (roots in (Thrun, 1998; Fei-Fei et al., 2006)). Dvornik et al. (2019) uses ensemble procedures for few-shot classifi-

cation. As to detection, Chen et al. (2018) proposes a regularized fine-tuning approach to transfer knowledge from a pre-trained detector to a few-shot detector. Schwartz et al. (2019) exploits metric-learning for modeling multi-modal distributions for each class. State-of-the-art few-shot detectors are usually serial (Wang et al., 2019; Yan et al., 2019; Wang et al., 2020; Fan et al., 2020; Wu et al., 2020; Xiao & Marlet, 2020). The existing literature can be seen as variations on a standard framework, where one splits data into two sets of categories: base classes $C_b$ (which have many training examples) and novel classes $C_n$ (which have few). The RPN and classifier are then trained with instances from the base classes, producing a detector for $|C_b|$ categories. The final layer of the resulting classifier is expanded to classify into $|C_b| + |C_n|$ classes by inserting random weights connecting the final feature layer to the $|C_n|$ novel categories. Now the model is fine-tuned using either only the novel class instances or a balanced dataset containing training instances of both base and novel classes. Wang et al. (2020) shows that a simple two-stage fine-tuning approach outperforms other complex methods. Much work seeks improvements by applying few-shot classification techniques. Kang et al. (2019) designs a meta-model that learns to reweight pre-trained features given few-shot data. Wang et al. (2019) and Yan et al. (2019) further explore the meta-learning direction by attaching meta-learned classifiers to Faster R-CNN. Wu et al. (2020) improves few-shot detection by positive sample refinement.

Relatively little work adjusts the proposal process, which is usually seen as robust to few-shot issues because there are many base examples. One possibility is to introduce attention mechanisms and feed category-aware features instead of plain image features into the proposal process (Hsieh et al., 2019; Fan et al., 2020; Xiao & Marlet, 2020; Osokin et al., 2020), as well as re-ranking proposals based on similarity with query images (Hsieh et al., 2019; Fan et al., 2020). Making the RPN category-aware improves the quality of novel class proposals, *but at inference time the model suffers from catastrophic forgetting of base categories* – current category-aware features cannot summarize the very large number of base class examples efficiently or accurately. An RPN that is generally well-behaved can still create serious trouble in the few-shot case by missing high IOU proposals for the novel classes during fine-tuning – the *proposal neglect* effect. We show that this problem is severe in the few-shot regime, and can be fixed by a carefully constructed ensemble of RPNs without substantial loss of performance for the base classes.

**Evaluating Few-Shot Detectors.** The standard procedure is to compute average precision (AP) separately for novel and base categories for a detector that is engineered to detect $|C_b| + |C_n|$ classes, typically using standard test/train splits and standard novel/base splits (Wang et al., 2020). This evaluation procedure is the same as in incremental few-shot detection (Yang et al., 2020; Pérez-Rúa et al., 2020). This procedure makes sense, because in most applications an incoming test image could contain instances from both base and novel classes. Furthermore, the standard procedure exposes any catastrophic forgetting that occurs. However, other evaluation methodologies occur, and some detectors are evaluated using variant procedures, making the comparison of AP's difficult. In one variant, the detector detects only the $|C_n|$ novel classes or only one novel class. In this paper, all reported results are for the standard procedure; when relevant, we re-implement and re-evaluate using the standard procedure.

## 3 OUR APPROACH

We believe that the proposal neglect effect is generic, and it applies to any detector that uses a structure like the standard structure. For this reason, we focus on finding and fixing the effect within a standard state-of-the-art few-shot object detection framework, as described below.

**Few-Shot Object Detection Framework.** We use the few-shot detection setting introduced in Kang et al. (2019). We split the dataset into two sets of categories: base classes $C_b$ and novel classes $C_n$. As shown in Figure 2, the training process is two-phase: (1) base classes training, and (2) fine-tuning with novel classes. In phase 1, the model is trained with base class instances which results in a $|C_b|$-way detector. After base classes training, weights for novel classes are randomly initialized, making the classifier a $(|C_b| + |C_n|)$-way classifier. In phase 2, the model is fine-tuned using either a set of few novel class instances or a balanced dataset containing both novel and base classes. After the fine-tuning phase, we evaluate our model by average precision (AP) on novel and base categories. Although the focus of few-shot detection is the novel classes, since most test images

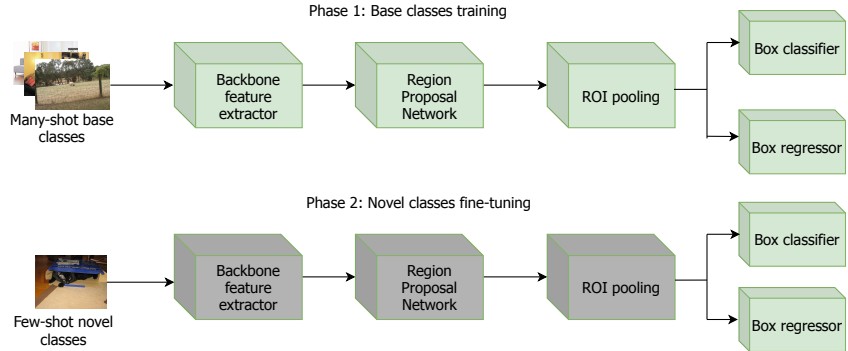

Figure 2: Illustration of two-phase training in Faster R-CNN. During phase 1, all blocks are trained. During phase 2, only the top-layer classifier and bounding box regressor are fine-tuned.

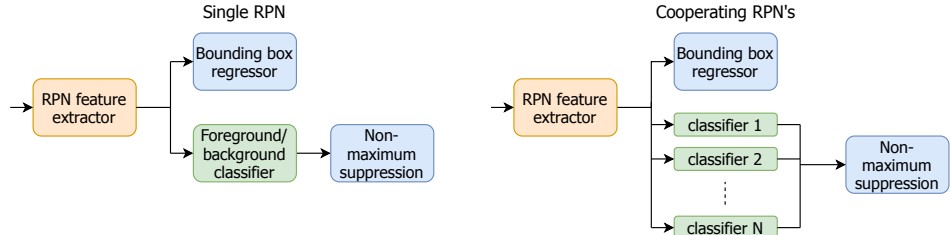

Figure 3: **Left** is the original structure for the RPN in Faster R-CNN; **right** is our CoRPNs, consisting of cooperating box classifiers.

contain instances from both base and novel classes, it is essential to maintain good performance on base classes.

We adopt the widely-used Faster R-CNN (Ren et al., 2015) as our base model. As shown in Figure 2, Faster R-CNN is a serial detector, which consists of a backbone image feature extractor, a regional proposal network (RPN), followed by the region of interest (ROI) pooling layer, and a bounding box classifier and a bounding box regressor on top of the model. The RPN determines if a box is a foreground or a background box. Following the RPN is non-maximum suppression (NMS) which ranks and selects top proposal boxes. After passing the ROI pooling layer, the predictor's head classifies and localizes each box. In phase 1, the whole model is trained on many-shot base class instances. Phase 2 fine-tunes the head classifier and the bounding box regressor with novel class instances; everything before and including the proposal generator is frozen.

**Learning Cooperating RPN's (CoRPNs).** We wish to avoid high IOU boxes for novel classes being dropped by RPN when novel classes are trained. Our strategy is to train multiple redundant RPN's. The RPN's should be distinct, but cooperate – if one misses a high IOU box, we want another to find it. However, they should not be so distinct that the classifier is flooded with false positives.

As Figure 3 shows, Faster R-CNN's RPN consists of a feature extractor, a binary classifier (which decides whether a box is foreground or background), and a bounding box regressor (which is not relevant to our current purpose). There is no reason for our RPN's to use distinct sets of features, and we do not want to create problems with variance, so we construct redundant classifiers while keeping both the feature extractor and the bounding box regressor shared between all RPN's. In what follows, a reference to an RPN is actually a reference to the RPN classifier, unless otherwise noted. An RPN with a single classifier is trained with a cross-entropy loss $\mathcal{L}_{cls} = \mathcal{L}_{CE}$ and produces a single prediction. In our case, we train $N$ different binary classifiers simultaneously, and must determine (1) what prediction is made at test time and (2) what gradient goes to what classifier at training time. At test time, a given box gets the score from the most certain RPN. If the highest foreground probability is closer to one than the highest background probability, the box is foreground; otherwise, it is background.

Training time is more interesting. Merely taking the gradient of the best RPN score is not good enough, because we may find that one RPN scores all boxes, and the others do nothing interesting. For any foreground box, we want at least one RPN to have a very strong foreground score *and* all others to have good foreground scores too (so that no foreground box is missed).

We use the following strategy. For a specific anchor box $i$, each RPN $j$ outputs a raw score $r_i^j$, indicating if the box is a foreground box or not: $r_i = [r_i^1, r_i^2, \ldots, r_i^N]$. After applying a sigmoid, the $j$th RPN produces the foreground probability $f_i^j = \sigma(r_i^j)$ for anchor box $i$. We choose the score from the $j^*$th RPN such that $j^* = \operatorname{argmin}_j \min\{f_i^j, 1 - f_i^j\}$, namely the most certain RPN which produces probability closest to the edge of the $[0, 1]$ interval. At training time, only the chosen $j^*$th RPN gets the gradient from anchor box $i$. The RPN selection procedure is per-box, even adjacent boxes can pass through different RPN's.

Other than the standard cross-entropy loss, we use two additional loss terms: a diversity loss $L_{div}$ encourages RPN's to be distinct, and a cooperation loss $L_{coop}$ encourages cooperation and suppresses foreground false negatives. The final loss $\mathcal{L}_{cls} := \mathcal{L}_{CE}^{j^*} + \lambda_d \mathcal{L}_{div} + \lambda_c \mathcal{L}_{coop}$, where $\lambda_d$ and $\lambda_c$ are trade-off hyperparameters.

**Enforcing Diversity.** We do not want our RPN's to be too similar. For each positive anchor box, RPN responses should be different because we want ensure that at novel class training time, if one RPN misses a high IOU anchor box, another will find it. To this end, we propose a methodology to enforce diversity among RPN's. Given a set of $N_A$ anchor boxes, the $N$ RPN's produce an $N$ by $N_A$ matrix of probabilities $\mathcal{F} = [f^1, f^2, \ldots, f^N]^T$. The covariance matrix $\Sigma(\mathcal{F})$ is $\Sigma(\mathcal{F}) = \mathrm{E}[(f^j - \mathrm{E}[f^j])(f^k - \mathrm{E}[f^k])]$. We define the diversity loss $\mathcal{L}_{div}$ by the log determinant loss $\mathcal{L}_{div} := -\log(\det(\Sigma(\mathcal{F})))$.

By the diversity loss, we encourage the probability matrix to have rank $N$, so each RPN is reacting differently on the collection of $N_A$ boxes. This procedure ensures each RPN to be the most certain RPN for some boxes, so that every RPN is being selected and trained. Omitting this loss can cause an RPN to receive no or little training.

**Learning to Cooperate.** We also want the RPN's to cooperate so that they all agree to a certain extent for foreground boxes. We propose a cooperation loss to prevent any RPN from firmly rejecting any foreground box. For foreground box $i$, with the $j$th RPN, we define the cooperation loss $\mathcal{L}_{coop}^{i,j} := \max\{0, \phi - f_i^j\}$, where $\phi$ is a constant parameter (usually less than 0.5), acting as a lower bound for each RPN's probability assigning to a foreground box. If a RPN's response is below $\phi$, that RPN is going to be penalized. The final cooperation loss is an average of cooperation losses over all foreground boxes and all RPN's.

## 4 EXPERIMENTS

**Datasets and Implementation Details.** We evaluate our approach on two widely-used few-shot detection benchmarks: MS-COCO (Lin et al., 2014) and PASCAL VOC (07 + 12) (Everingham et al., 2010). For a fair comparison, we use the same train/test splits and novel class instances as in Kang et al. (2019); Wang et al. (2020) to train and evaluate all models. On COCO, we report base/novel classes AP, AP50, and AP75 under shots 1, 2, 3, 5, 10, and 30. On PASCAL VOC, we report AP50 for three different base/novel classes splits under shots 1, 2, 3, 5, and 10. Consistent with recent works (Yan et al., 2019; Wang et al., 2020), we mainly use an ImageNet pre-trained (Russakovsky et al., 2015) ResNet-101 architecture (He et al., 2016) as the backbone, unless otherwise noted. Following Wang et al. (2020), for all TFA and CoRPNs results, ground-truth boxes are appended as training samples. As reported in Wang et al. (2020), adding the ground-truth boxes leads to a 0.5% AP gain on COCO. More implementation details can be found in the Appendix.

**Hyperparameters on PASCAL VOC.** For ease of comparison, we use the same values of all shared training and fine-tuning hyperparameters (batch size, learning rate, momentum, weight decay, etc.) as Wang et al. (2020). CoRPNs have the following additional hyperparameters: the number of RPN's, the cooperation loss threshold $\phi$, the diversity loss trade-off $\lambda_d$, and the cooperation loss trade-off $\lambda_c$. The number of RPN's does not affect the results much as long as other hyperparameters are suitably selected (Figure 4). For results described here, we use the following hyperparameters sets: VOC split 1, 5 RPNs, $\phi = 0.2$, $\lambda_c = 2$, $\lambda_d = 0.025$; VOC split 2, 5 RPNs, $\phi = 0.1$, $\lambda_c = 1$, $\lambda_d$

= 0.025; VOC split 3, 2 RPNs, $\phi$ = 0.5, $\lambda_c$ = 1, $\lambda_d$ = 0.3 (more information on good ranges in the Appendix). We selected hyperparameters using base classes and the following criteria on PASCAL VOC: (1) The average number of RPNs responding positively to a foreground box (a larger average means that the RPN's are cooperating well with each other by responding positively to foreground boxes); (2) The cumulative variance in the RPN's response matrix on foreground boxes (details in Appendix); (3) The average number of positive samples (larger means that the CoRPNs produce more high IOU boxes). *Because the hyperparameters are selected on base classes, we can use a different set per split without concerns about generalization.*

**Hyperparameters on COCO.** For COCO, we selected hyperparameters that worked well on all three novel splits in PASCAL VOC and used those. This means that improvements in COCO are not caused by hyperparameter variance. Results reported were obtained with 5 RPNs, $\phi$ = 0.3, $\lambda_c$ = 1, $\lambda_d$ = 0.025. We report results for two other hyperparameter sets in the Appendix.

**Baselines and Evaluation Procedure.** To investigate the *proposal neglect* effect and for a fair comparison, we mainly focus on comparing against the state-of-the-art baseline TFA (Wang et al., 2020). Our approach introduces CoRPNs into TFA, while keeping other model components and design choices unchanged. In addition, we thoroughly compare with a variety of recent few-shot detectors, including CoAE (Hsieh et al., 2019), Meta R-CNN (Yan et al., 2019), FSOD (Fan et al., 2020), MPSR (Wu et al., 2020), FSDetView (Xiao & Marlet, 2020), and ONCE (Pérez-Rúa et al., 2020). These baselines address other aspects of few-shot detection which are different from us (Section 2), and their modifications are thus largely orthogonal to our effort here. Note that our evaluation follows the standard procedure in Wang et al. (2020). As explained in Section 2, this standard procedure computes AP separately for novel and base categories for a detector that is engineered to detect $|C_b| + |C_n|$ classes. For a fair comparison, we re-evaluate CoAE (Hsieh et al., 2019) and FSOD (Fan et al., 2020) using the standard procedure.

**Main Results.** Tables 1 and 2 summarize the results for novel classes on COCO and PASCAL VOC, respectively, and Table 4 summarizes the results for base classes on COCO.

***Comparisons with the Main Baseline TFA on Novel Classes.*** From Tables 1 and 2, we have the following important observations. (I) Our approach produces a substantial improvement in AP over TFA (Wang et al., 2020) on novel classes *in the very low-shot regime* (1, 2, and 3 shots), and marginal improvement or sometimes slight degradation in the higher-shot regime. These improvements are manifest for both existing benchmarks. Interestingly, on the more challenging COCO dataset, our improvements over TFA are *consistent across different shots* (except slight degradation in 30 shot under AP75 with fully-connected classifier). (II) We investigate two types of classifiers: either fully-connected (denoted as 'fc' in Tables 1 and 2) or cosine (denoted as 'cos' in Tables 1 and 2). Note that our approach obtains improvements regardless of classifier choice. This is because CoRPNs is a strategy to control variance in the estimate of classifier parameters *that applies independently of the classifier*. Any high IOU box missing from the RPN output in the training phase must cause variance for the few-shot regime. Because there are very few such boxes, the effect of not losing boxes is pronounced. We provide visualization comparisons of detection results in the Appendix.

***Is Proposal Neglect Occurring?*** Of course, CoRPNs might outperform TFA for some obscure reason. But there is good evidence that proposal neglect occurs, and that CoRPNs help. Table 3 records the average number of foreground boxes at three IoU's produced by CoRPNs and by TFA at the fine-tuning stage for novel classes. This table shows that CoRPNs produce more (and better) foreground boxes than TFA. We expect these small improvements in the box proposal pool to have a large effect, because such boxes improve the classifier's model of appearance variation for novel classes; small changes in this model can result in significant movements of the class boundaries.

***Comparisons with Other State-of-the-Art Approaches.*** With our simple modification on RPN, we also outperform other sophisticated approaches on both benchmarks in the very low-shot regime, and achieve comparable performance in the higher-shot regime. In particular, we significantly outperform those baselines that introduce attention mechanisms for adjusting proposal generation (Hsieh et al., 2019; Fan et al., 2020). For other approaches that improve few-shot detection from different perspectives, such as exploiting better multi-scale representation (Wu et al., 2020), our approach can be potentially combined with them for further improvements.

***Comparisons on Base Classes.*** While improving detection on novel classes through fine-tuning, we maintain strong performance on base classes *without suffering from catastrophic forgetting* as

shown in Table 4. By contrast, the performance of the state-of-the-art baselines dramatically drops, demonstrating that they cannot simultaneously deal with both novel and base classes.

| | Method | Backbone | 1-shot AP | AP50 | AP75 | 2-shot AP | AP50 | AP75 | 3-shot AP | AP50 | AP75 |
|---|---|---|---|---|---|---|---|---|---|---|---|
| Ours | CoRPNs w/ fc | ResNet-101 | 3.4 | 6.7 | 3.0 | 5.4 | 10.4 | 5.1 | 7.1 | 13.7 | 6.8 |
| | CoRPNs w/ cos | ResNet-101 | 4.1 | 7.2 | 4.4 | 5.4 | 9.6 | 5.6 | 7.1 | 13.2 | 7.2 |
| | CoRPNs w/ cos | ResNet-50 | 3.7 | 6.8 | 3.8 | 4.7 | 8.8 | 4.4 | 6.3 | 12.0 | 6.0 |
| Main baselines | TFA w/ fc (Wang et al., 2020) | ResNet-101 | 2.9 | 5.7 | 2.8 | 4.3 | 8.5 | 4.1 | 6.7 | 12.6 | 6.6 |
| | TFA w/ cos (Wang et al., 2020) | ResNet-101 | 3.4 | 5.8 | 3.8 | 4.6 | 8.3 | 4.8 | 6.6 | 12.1 | 6.5 |
| Other baselines | Meta R-CNN (Yan et al., 2019) | ResNet-101 | – | – | – | – | – | – | – | – | – |
| | FRCN+ft-full (Wang et al., 2020) | ResNet-101 | – | – | – | – | – | – | – | – | – |
| | MPSR (Wu et al., 2020) | ResNet-101 | 2.3 | 4.1 | 2.3 | 3.5 | 6.3 | 3.4 | 5.2 | 9.5 | 5.1 |
| | FsDetView (Xiao & Marlet, 2020) | ResNet-101 | 2.9 | 8.3 | 1.2 | 3.7 | 10.3 | 1.6 | 4.7 | 12.9 | 2.0 |
| | ONCE (Pérez-Rúa et al., 2020) | ResNet-50 | 0.7 | – | – | – | – | – | – | – | – |
| | FSOD* (Fan et al., 2020) | ResNet-50 | 2.4 | 4.8 | 2.0 | 2.9 | 5.9 | 2.7 | 3.7 | 7.2 | 3.3 |
| | Method | Backbone | 5-shot AP | AP50 | AP75 | 10-shot AP | AP50 | AP75 | 30-shot AP | AP50 | AP75 |
| Ours | CoRPNs w/ fc | ResNet-101 | 8.9 | 16.9 | 8.6 | 10.5 | 20.2 | 9.8 | 13.5 | 25.0 | 12.9 |
| | CoRPNs w/ cos | ResNet-101 | 8.8 | 16.4 | 8.7 | 10.6 | 19.9 | 10.1 | 13.9 | 25.1 | 13.9 |
| | CoRPNs w/ cos | ResNet-50 | 7.8 | 14.4 | 7.6 | 9.0 | 17.6 | 8.3 | 13.4 | 24.6 | 13.3 |
| Main baselines | TFA w/ fc (Wang et al., 2020) | ResNet-101 | 8.4 | 16.0 | 8.4 | 10.0 | 19.2 | 9.2 | 13.4 | 24.7 | 13.2 |
| | TFA w/ cos (Wang et al., 2020) | ResNet-101 | 8.3 | 15.3 | 8.0 | 10.0 | 19.1 | 9.3 | 13.7 | 24.9 | 13.4 |
| Other baselines | Meta R-CNN (Yan et al., 2019) | ResNet-101 | – | – | – | 8.7 | – | 6.6 | 12.4 | – | 10.8 |
| | FRCN+ft-full (Wang et al., 2020) | ResNet-101 | – | – | – | 9.2 | – | 9.2 | 12.5 | – | 12.0 |
| | MPSR (Wu et al., 2020) | ResNet-101 | 6.7 | 12.6 | 6.4 | 9.7 | 18.0 | 9.4 | 13.7 | 25.0 | 13.4 |
| | FsDetView (Xiao & Marlet, 2020) | ResNet-101 | 5.8 | 15.6 | 2.9 | 6.7 | 17.3 | 3.7 | 9.6 | 22.1 | 6.6 |
| | ONCE (Pérez-Rúa et al., 2020) | ResNet-50 | 1.0 | – | – | 1.2 | – | – | – | – | – |
| | FSOD* (Fan et al., 2020) | ResNet-50 | 4.2 | 8.2 | 4.0 | 4.3 | 8.7 | 3.8 | 5.4 | 10.4 | 5.0 |

Table 1: Few-shot detection performance on COCO novel classes. The upper row shows the 1, 2, 3-shot results, and the lower row shows the 5, 10, 30-shot results. Results in red are the best, and results in blue are the second best. All approaches are evaluated following the standard procedure in Wang et al. (2020). *Model re-evaluated using the standard procedure for a fair comparison. '−' denotes that numbers are not reported in the corresponding paper. Note that the publicly released models of ONCE and FSOD are based on ResNet-50; we include our CoRPNs based on ResNet-50 as well for a fair comparison. CoRPNs consistently outperform state of the art in almost all settings, *with substantial improvements especially in the very few-shot regime*. Our strategy is also effective *regardless of classifier choice*.

| | Method | Novel Set 1 shot=1 | 2 | 3 | 5 | 10 | Novel Set 2 shot=1 | 2 | 3 | 5 | 10 | Novel Set 3 shot=1 | 2 | 3 | 5 | 10 |
|---|---|---|---|---|---|---|---|---|---|---|---|---|---|---|---|---|
| Ours | CoRPNs w/ fc (Ours) | 40.8 | 44.8 | 45.7 | 53.1 | 54.8 | 20.4 | 29.2 | 36.3 | 36.5 | 41.5 | 29.4 | 40.4 | 44.7 | 51.7 | 49.9 |
| | CoRPNs w/ cos (Ours) | 44.4 | 38.5 | 46.4 | 54.1 | 55.7 | 25.7 | 29.5 | 37.3 | 36.2 | 41.3 | 35.8 | 41.8 | 44.6 | 51.6 | 49.6 |
| Main baselines | TFA w/ fc (baseline) (Wang et al., 2020) | 36.8 | 29.1 | 43.6 | 55.7 | 57.0 | 18.2 | 29.0 | 33.4 | 35.5 | 39.0 | 27.7 | 33.6 | 42.5 | 48.7 | 50.2 |
| | TFA w/ cos (baseline) (Wang et al., 2020) | 39.8 | 36.1 | 44.7 | 55.7 | 56.0 | 23.5 | 26.9 | 34.1 | 35.1 | 39.1 | 30.8 | 34.8 | 42.8 | 49.5 | 49.8 |
| Other baselines | FRCN+ft-full (Wang et al., 2020) | 15.2 | 20.3 | 29.0 | 40.1 | 45.5 | 13.4 | 20.6 | 28.6 | 32.4 | 38.8 | 19.6 | 20.8 | 28.7 | 42.2 | 42.1 |
| | Meta R-CNN (Yan et al., 2019) | 19.9 | 25.5 | 35.0 | 45.7 | 51.5 | 10.4 | 19.4 | 29.6 | 34.8 | 45.4 | 14.3 | 18.2 | 27.5 | 41.2 | 48.1 |
| | CoAE* (Hsieh et al., 2019) | 12.7 | 14.6 | 14.8 | 18.2 | 21.7 | 4.4 | 11.3 | 20.5 | 18.0 | 19.0 | 6.3 | 7.6 | 9.5 | 15.0 | 19.0 |
| | MPSR (Wu et al., 2020) | 41.7 | 43.1 | 51.4 | 55.2 | 61.8 | 24.4 | 29.5 | 39.2 | 39.9 | 47.8 | 35.6 | 40.6 | 42.3 | 48.0 | 49.7 |
| | FsDetView (Xiao & Marlet, 2020) | 24.2 | 35.3 | 42.2 | 49.1 | 57.4 | 21.6 | 24.6 | 31.9 | 37.0 | 45.7 | 21.2 | 30.0 | 37.2 | 43.8 | 49.6 |

Table 2: Few-shot detection performance (AP50) on PASCAL VOC novel classes under three base/novel splits. All models are based on Faster R-CNN with a ResNet-101 backbone. Results in red are the best, and results in blue are the second best. We follow the standard evaluation procedure in Wang et al. (2020). *Model re-evaluated under the standard procedure. CoRPNs outperform all the baselines in the very low shots, and achieve comparable performance in the higher shots.

| | VOC Split 1 | | | VOC Split 2 | | |
|---|---|---|---|---|---|---|
| Method | Avg #FG (0.5) | Avg #FG (0.6) | Avg #FG (0.7) | Avg #FG (0.5) | Avg #FG (0.6) | Avg #FG (0.7) |
| CoRPNs | 18.67 | 7.85 | 2.54 | 14.14 | 6.27 | 2.06 |
| TFA (Wang et al., 2020) | 17.61 | 7.41 | 2.37 | 13.61 | 5.86 | 2.03 |
| | VOC Split 3 | | | COCO | | |
| Method | Avg #FG (0.5) | Avg #FG (0.6) | Avg #FG (0.7) | Avg #FG (0.5) | Avg #FG (0.6) | Avg #FG (0.7) |
| CoRPNs | 16.02 | 6.65 | 2.30 | 11.97 | 5.41 | 1.84 |
| TFA (Wang et al., 2020) | 16.76 | 6.75 | 2.13 | 11.55 | 5.32 | 1.79 |

Table 3: CoRPNs produce more and better boxes at the fine-tuning stage. This table shows the average number of foreground boxes TFA and CoRPNs produce for novel classes in the 1-shot experiments. 'Avg # FG' is calculated from proposals after non-maximum suppression. We exclude the ground truth box, which the classifier always sees, in calculating 'Avg # FG'. At each of three IoU thresholds, for two of three splits in PASCAL VOC and for COCO, CoRPNs reliably produce more boxes. This table uses the same hyperparameters in Tables 1 and 2. The relative improvement is generally large; we expect small improvements to have large effects, because the classifier must use the very few boxes it receives to build a model of variation in object appearance.

| | Method | Backbone | 1-shot finetuned | | | 2-shot finetuned | | | 3-shot finetuned | | |
|---|---|---|---|---|---|---|---|---|---|---|---|
| | | | AP | AP50 | AP75 | AP | AP50 | AP75 | AP | AP50 | AP75 |
| Ours | CoRPNs w/ cos | ResNet-101 | 34.1 | 55.1 | 36.5 | 34.7 | 55.3 | 37.3 | 34.8 | 55.2 | 37.6 |
| | CoRPNs w/ cos | ResNet-50 | 32.1 | 52.9 | 34.4 | 32.7 | 52.9 | 35.5 | 32.6 | 52.4 | 35.4 |
| Main baseline | TFA w/ cos (Wang et al., 2020) | ResNet-101 | 34.1 | 54.7 | 36.4 | 34.7 | 55.1 | 37.6 | 34.7 | 54.8 | 37.9 |
| | MPSR (Wu et al., 2020) | ResNet-101 | 12.1 | 17.1 | 14.2 | 14.4 | 20.7 | 16.9 | 15.8 | 23.3 | 18.3 |
| Other baselines | FsDetView (Xiao & Marlet, 2020) | ResNet-101 | 1.9 | 5.7 | 0.8 | 2.7 | 8.2 | 0.9 | 3.9 | 10.8 | 2.0 |
| | FSOD* (Fan et al., 2020) | ResNet-50 | 11.9 | 20.3 | 12.5 | 15.6 | 24.4 | 17.2 | 17.4 | 27.3 | 19.0 |
| | Method | Backbone | 5-shot finetuned | | | 10-shot finetuned | | | 30-shot finetuned | | |
| | | | AP | AP50 | AP75 | AP | AP50 | AP75 | AP | AP50 | AP75 |
| Ours | CoRPNs w/ cos | ResNet-101 | 34.7 | 54.8 | 37.5 | 34.6 | 54.5 | 38.2 | 35.8 | 55.4 | 39.4 |
| | CoRPNs w/ cos | ResNet-50 | 32.3 | 51.6 | 34.9 | 32.7 | 51.9 | 36.0 | 33.5 | 52.7 | 37.0 |
| Main baseline | TFA w/ cos (Wang et al., 2020) | ResNet-101 | 34.7 | 54.4 | 37.6 | 35.0 | 55.0 | 38.3 | 35.8 | 55.5 | 39.4 |
| | MPSR (Wu et al., 2020) | ResNet-101 | 17.4 | 25.9 | 19.8 | 19.5 | 29.5 | 21.9 | 21.0 | 32.4 | 23.4 |
| Other baselines | FsDetView (Xiao & Marlet, 2020) | ResNet-101 | 5.3 | 14.2 | 2.8 | 6.4 | 15.9 | 4.1 | 9.0 | 20.6 | 6.7 |
| | FSOD* (Fan et al., 2020) | ResNet-50 | 16.7 | 26.2 | 18.3 | 18.9 | 29.3 | 20.7 | 18.8 | 29.4 | 20.1 |

Table 4: Detection performance on COCO base classes after *fine-tuning with k-shot novel classes instances.* *Model re-evaluated under the standard procedure (Wang et al., 2020). Our CoRPNs and TFA (Wang et al., 2020) maintain good performance on base classes, whereas MPSR (Wu et al., 2020) and FsDetView (Xiao & Marlet, 2020) suffer from severe catastrophic forgetting.

| Method | AP50 |
|---|---|
| TFA (Wang et al., 2020) | 30.8 |
| 2 RPN's CoRPNs | 35.8 |
| 2 RPN's Diversity Loss Only | 28.4 |
| 2 RPN's Cooperation Loss Only | 27.2 |

Table 5: Our diversity loss and cooperation loss are both required for CoRPNs to obtain an improvement. The table presents novel class AP50 of CoRPNs with diversity or cooperation loss only, under PASCAL VOC split 3, shot 1.

| Method | AP50 |
|---|---|
| 2 RPN's, Dvornik et al. (2019) | 32.4 |
| 2 RPN's, Bootstrapping | 31.8 |
| 2 RPN's, CoRPNs (Ours) | **35.8** |

Table 6: Our diversity term – the log-determinant loss – offers improvements over the pairwise cosine similarity based diversity loss in Dvornik et al. (2019), and a bootstrapping baseline. The table shows novel class AP50 of all models under PASCAL VOC novel split 3, shot 1.

**Ablation Studies.** We conduct a series of ablations to evaluate the contribution of each component and different design choices. Specifically, we show that (1) with our novel loss terms, CoRPNs outperform naive RPN ensembles; (2) CoRPNs outperform a bootstrapping baseline; (3) CoRPNs outperform baselines with larger RPN sub-networks; (4) CoRPNs outperform an existing cosine loss based diversity method; (5) our cooperation loss is helpful; (6) the number of RPN's does not matter very much; (7) CoRPNs need both the diversity loss and the cooperation loss.

***CoRPNs vs. Naive RPN Ensembles.*** We compare CoRPNs with an ensemble of the same number of RPN's trained separately, with different initializations. At the fine-tuning phase, we apply the same RPN selection mechanism to both approaches. Figure 4 shows that CoRPNs (middle, with selected hyperparameters) outperforms naive ensembles (right) across different numbers. This suggests that *pure redundancy does not lead to much improvement*, confirming that our loss terms enforce diversity and cooperation. Additionally, since each model in naive ensemble is trained separately, the number of parameters and training time of our CoRPNs are $N$ times less than naive ensembles.

***CoRPNs vs. Bootstrapping.*** We compare CoRPNs with a bootstrapping baseline in Table 6, with implementation details include in Appendix. CoRPNs outperform the bootstrapping baseline, suggesting that *RPN variance obtained by training data variation does not lead to much improvement.*

***CoRPNs vs. Larger RPN.*** We compare CoRPNs with baselines having larger RPN sub-networks in Table 7. We find that using larger RPN sub-networks does not improve performance, suggesting that the advantage of CoRPNs is *not simply the result of using more parameters*. Please find implementation details in the Appendix.

***CoRPNs vs. Cosine Loss Diversity.*** Table 6 confirms that in this context, our diversity loss – the log-determinant loss – outperforms another diversity loss introduced in Dvornik et al. (2019). Dvornik et al. (2019) encourage diversity among classifiers by enlarging the cosine distance in classifier outputs. We replace our diversity loss term by a pairwise cosine similarity loss among RPN's (for $N$ RPN's, the pairwise cosine similarity loss is an average over $N(N-1)/2$ pairs).

***Cooperation loss on Avoiding False Negatives.*** Our cooperation loss $\mathcal{L}_{coop}$ pushes all RPN's to agree on a certain degree for all foreground boxes by having a threshold hyperparameter $\phi$. Table 8 shows that $\mathcal{L}_{coop}$ reduces the number of false-negatives, and false-negatives decreases as $\phi$ increases.

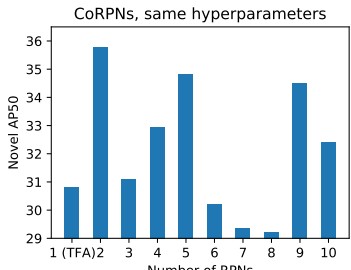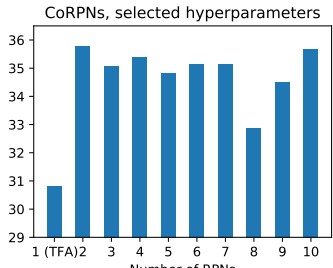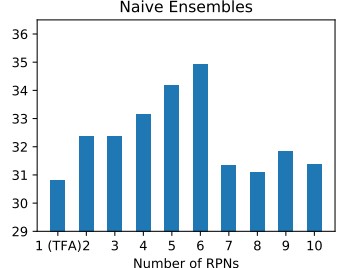

Figure 4: Effect of the number of RPN's on novel AP50, under PASCAL VOC novel split 3, shot 1. The **left** shows results using same hyperparameters ($\phi$, $\lambda_d$, and $\lambda_c$). There is considerable variance across the number of RPN's. The **middle** shows the results of selected hyperparameters (with selection criteria discussed earlier in this section) for different numbers of RPN's. CoRPNs with any number of RPN's outperform TFA (number of RPN = 1), and the variance is much smaller than on the **left**. The **right** shows naive ensemble results. Naive ensemble improves the performance but not as good as CoRPNs with selected hyperparameters (middle).

| Method | #param added | AP50 |
|---|---|---|
| TFA, original | 0 | 30.8 |
| TFA, Large RPN | 28*256 | 25.9 |
| TFA, Larger RPN | 268*256 | 27.8 |
| CoRPNs, 2 RPN's | 3*256 | **35.8** |
| CoRPNs, 5 RPN's | 12*256 | **34.8** |
| CoRPNs, 10 RPN's | 27*256 | **35.7** |

Table 7: This table shows novel class AP50 of TFA (Wang et al., 2020) with a large RPN and an even larger RPN, and compares with CoRPNs with different numbers of RPN's. All results are with PASCAL VOC novel split 3, shot 1. The second column presents how many additional parameters are introduced to the original TFA model. Using larger RPN sub-networks does not improve the performance, while CoRPNs significantly improve with fewer added parameters.

| Method | | AP50 | Avg # FN ($\downarrow$) | Avg # FG ($\uparrow$) |
|---|---|---|---|---|
| TFA (Wang et al., 2020) | | 28.9 | 3.1 | 17.6 |
| CoRPNs w/ | $\phi = 0.1$ | **29.5** | 2.4 | 21.3 |
| | $\phi = 0.3$ | **31.5** | 3.0 | 18.5 |
| | $\phi = 0.5$ | **32.2** | 2.5 | 18.3 |
| | $\phi = 0.7$ | 26.8 | 1.3 | 20.3 |
| | $\phi = 0.9$ | **31.7** | 0.8 | 19.0 |

Table 8: Our threshold $\phi$ controls the average number of false-negative foreground boxes and the number of foreground samples. 'Avg # FN' is the average number of novel class training foreground boxes misclassified by the RPN classifier (calculated before non-maximum suppression). 'Avg # FG' is the same as 'Avg # FG (0.5)' in Table 3, which is the average number of foreground boxes after non-maximum suppression. When $\phi$ gets larger, all RPN's produce higher scores for all foreground boxes, it is more and more unlikely that CoRPNs get foreground boxes wrong, so 'Avg # FN' decreases. CoRPNs with all $\phi$ values also have larger 'Avg # FG' compared with TFA. All results are under PASCAL VOC novel split 1, shot 1. Different from other experiments, here we fine-tune with novel classes only (detector is $C_n$-way).

**The Number of RPN's.** We present results with different numbers of RPN's in Figure 4. We find that the number of RPN's doesn't affect the result much with selected hyperparameters.

**CoRPNs need both diversity and cooperation losses.** Table 5 shows that using either loss alone does not improve the performance.

## 5 CONCLUSION

A substantial improvement in few-shot performance can be obtained by engineering a system of RPN's to ensure that high IOU boxes for few-shot training images almost always pass the system of RPN's. Our method achieves a new state of the art on widely-used benchmarks and outperforms the current state of the art by a large margin in the very few-shot regime. This is because, as ablation experiments indicate, proposal neglect is a real effect. We engineered our ensemble of RPN's by using a diversity loss and a cooperation loss, which gave us some control of the false negative rate. We showed that controlling proposal neglect is an essential part of building a strong few-shot object detector by showing that the state-of-the-art few-shot detector could be improved in this way. We do not claim that our method is the best way to control proposal neglect and we plan to investigate methods that can offer more delicate control. Furthermore, it is very likely that proposal neglect applies to one-stage detectors like YOLO (Redmon & Farhadi, 2018), and future work will investigate this.

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

# A  IMPLEMENTATION DETAILS

Following Wang et al. (2020), we use Faster R-CNN as our base model and use an ImageNet pre-trained (Russakovsky et al., 2015) ResNet-101 with a Feature Pyramid Network (Lin et al., 2017) as the backbone.

As discussed in the main paper, the training process of CoRPNs is two-phase. Our training and fine-tuning procedures are consistent with Wang et al. (2020). On PASCAL VOC, at phase 1 base classes training, each model is trained on the union set of VOC 07+12 trainval data. Evaluation is on the VOC 07 test set. At the fine-tuning phase, each model is fine-tuned with a balanced few-shot dataset sampled from VOC 07+12 that contains both base classes and novel classes. On COCO, the fine-tuning phase is two-stage: at stage 1, we fine-tune the model on novel classes; at stage 2, we then fine-tune the model with a balanced few-shot dataset containing both base and novel classes.

We showed the effect of using different levels of $\phi$ in ablation studies of the main paper. As discussed in the main paper, the number of RPN's doesn't affect the results much with selected hyperparameters. For 5 RPN's, we find that $\phi \in [0.1, 0.3]$ works best, and results are relatively stable for $\lambda_c \in [0.5, 2]$ and $\lambda_d \in [0.01, 0.1]$.

***Evaluation.*** As mentioned in the main paper, we use the standard evaluation procedure for all the compared models for a fair comparison. We also compare against other approaches with the same novel classes instances and test images. On COCO, We fine-tuned and re-evaluated the publicly released models of MPSR (Wu et al., 2020) and FSDetView (Xiao & Marlet, 2020) with the same novel classes instances in Wang et al. (2020). In the standard evaluation procedure, when a test image comes in, the model has no assumption on what categories the image contains. We re-evaluated FSOD (Fan et al., 2020) and CoAE (Hsieh et al., 2019) using the standard evaluation. In Wang et al. (2020) and CoRPNs, the classifier is a $(|C_b| + |C_n|)$-way classifier. FSOD replaces it with a class-agnostic 2-way classifier, which only determines if an object is foreground or background. At inference time, FSOD produces a balanced number of proposals per novel category and concatenates these proposals before NMS. In the standard evaluation, we produce a balanced number of proposals for every category, including base categories. In CoAE, for a test image containing a certain category, CoAE samples support image(s) from this category and collects boxes based on the support image(s). We modify the inference process by providing each test image support image(s) from each category and performing a forward pass for all categories. We then collect boxes from all categories and evaluate them. For a fair comparison, for the results reported in the main paper, we also fine-tuned FSOD and CoAE with the same novel category instance(s) as in Wang et al. (2020) and CoRPNs.

***Training Time and Memory.*** The only architectural difference between CoRPNs and TFA (Wang et al., 2020) is that CoRPNs have multiple RPN classifiers. In TFA, the RPN's classifier is a 1*1 convolutional layer, with input channels as the number of feature channels and output channels as the number of cell anchors. In TFA, the input channels are 256, and the output channels are 3. CoRPNs with 5 RPN's add four additional RPN classifiers to the model, 256*12 additional parameters in total. In our experiments, we find CoRPNs with 5 RPN's increase the training time by 3%, with roughly the same memory compared with TFA.

***Selection Procedure – Cumulative Variance.*** We mentioned the hyperparameter selection criteria in the experiment section. The second criteria is the cumulative variance in the RPN's response matrix to foreground boxes. Given a set of $M$ anchor boxes, the $N$ RPN's produce an $N$ by $M$ matrix of probabilities $\mathcal{F} = [f^1, f^2, \ldots, f^N]^T$. We run a PCA on the $\mathcal{F}$ with $N$ components and compare the cumulative percentage of variance explained by each of the selected components. We would like the variance to be distributed across components.

***Baseline – Bootstrapping.*** We compare CoRPNs with a bootstrapping baseline in Table 6. We construct the bootstrapping baseline by using multiple RPN classifiers, same as CoRPNs. Instead of selecting the most certain RPN to get gradients, we randomly select an RPN to get gradients during training. In the fine-tuning stage and the inference time, we use the same selection procedure as CoRPNs (select the most certain RPN).

***Baseline – Larger RPN.*** We compare CoRPNs with two baselines with larger RPN sub-networks in Table 7. In CoRPNs, all RPN classifiers share the same RPN feature extractor, as shown in Figure 3. To have a larger RPN sub-network, we need to enlarge the feature dimensions. We tried

| | 1-shot | | | 2-shot | | | 3-shot | | |
|---|---|---|---|---|---|---|---|---|---|
| Method | AP | AP50 | AP75 | AP | AP50 | AP75 | AP | AP50 | AP75 |
| CoRPNs w/ cos, param set 1 (Table 1 reported) | 4.13 | 7.20 | 4.40 | 5.41 | 9.58 | 5.62 | 7.06 | 13.19 | 7.24 |
| CoRPNs w/ cos, param set 2 | 4.47 | 7.27 | 5.10 | 5.14 | 9.28 | 5.35 | 7.21 | 13.12 | 7.32 |
| CoRPNs w/ cos, param set 3 | 4.01 | 6.97 | 4.31 | 5.16 | 8.96 | 5.53 | 6.90 | 12.89 | 6.74 |
| CoRPNs w/ cos, Standard Deviation | 0.20 | 0.13 | 0.36 | 0.12 | 0.25 | 0.11 | 0.13 | 0.13 | 0.26 |
| CoRPNs w/ cos, 95% confidence intervals | 4.21±0.22 | 7.15±0.15 | 4.59±0.41 | 5.23±0.14 | 9.27±0.29 | 5.50±0.13 | 7.06±0.14 | 13.06±0.15 | 7.09±0.29 |
| TFA w/ cos (Wang et al., 2020) | 3.43 | 5.80 | 3.83 | 4.57 | 8.30 | 4.78 | 6.57 | 12.11 | 6.48 |

Table 9: We present COCO 1, 2, 3-shot results under three sets of hyperparameters. All hyperparameters are selected from PASCAL VOC by the selection criteria described in the experiment section. This table also shows the standard deviation and 95% confidence intervals across three sets of hyperparameters. All results from CoRPNs outperform TFA. Results with 95% confidence intervals also show that the improvements are significant.

two options: a large RPN where the RPN feature extractor's output channels increase from 256 to 272, and a larger RPN where the output channels double to 512. We also change the RPN classifier and bounding box regressor to take in larger features for both options.

## B   Performance Variance Across Hyperparameters

We provide the results from two other sets of hyperparameters on COCO and computed the mean, standard deviation, and 95% confidence intervals from the three sets of results. All three hyperparameter sets are selected from PASCAL VOC, by the criteria described in the experiments section. As shown in Table 9, the improvement by CoRPNs is significant.

## C   Results on Base classes

CoRPNs use a two-phase approach like TFA (Wang et al., 2020). Here we provide the base classes AP after phase 1 base classes training. As shown in Tables 10 and 11, our performance on base classes is comparable with TFA (Wang et al., 2020). Note that the setting of Tables 10 and 11 is different from that of Table 3 (COCO base classes) in the main paper, where the results are reported after phase 2 $k$-shot novel classes fine-tuning.

| Method | Split 1 | Split 2 | Split 3 |
|---|---|---|---|
| TFA w/ cos (Wang et al., 2020) | 80.8 | **81.9** | 82.0 |
| CoRPNs w/ cos (Ours) | **81.2** | 81.8 | **82.4** |

Table 10: Base classes AP50 on PASCAL VOC after phase 1 base class training. The same parameter setting applies to both models. Notice that ours is comparable to the results of TFA.

| Method | AP | AP50 | AP75 |
|---|---|---|---|
| TFA w/ cos (Wang et al., 2020) | **39.2** | 59.3 | **42.8** |
| CoRPNs w/ cos (Ours) | **39.2** | 59.4 | 42.5 |

Table 11: Base classes AP, AP50, and AP75 on COCO after phase 1 base class training. The same parameter setting applies to both models. Ours is comparable to the reported numbers of TFA.

## D   Baseline: Finetune RPN

Table 12 provides results with RPNs fine-tuned in stage 2. In addition to fine-tuning the bounding box regressor and the classifier, we also fine-tuned the RPN and the ROI feature extractor.

| Method | Novel Set 1, 1-shot | Novel Set 2, 1-shot | Novel Set 3, 1-shot |
|---|---|---|---|
| TFA (Wang et al., 2020) | 39.8 | 23.5 | 30.8 |
| TFA, RPN fine-tuned | 36.9 | 20.7 | 34.1 |
| CoRPNs | **44.4** | **25.7** | **35.8** |

Table 12: This table presents results from the RPN finetuned baseline, under PASCAL VOC three novel splits, all in 1-shot. Finetuning RPN improves the performance in one split but degrades the performance in two other splits. CoRPNs are the best in all cases.

with cosine classifiers on PASCAL three splits, all on 1-shot. We also need to fine-tune the box head (the feature extractor in the ROI head) after the RPN. We found the results are not comparable to the TFA baselines;

# E  ERROR ANALYSIS

We provide some proposal and final detection visualizations of CoRPNs and TFA (Wang et al., 2020) on PASCAL VOC. Specifically, Figure 5 presents the proposals and the detection results for cat test images from 1-shot fine-tuned models under base/novel split 3. Cat is a novel category for this split. Compare to TFA, CoRPNs generates more cat boxes. However, since the few-shot classifier is weak on novel classes, and the classifier has engaged with strong priors with base classes, cats are often misclassified as other base categories, leading to a performance drop. This suggests that if we have a better classifier, we can further improve the performance.

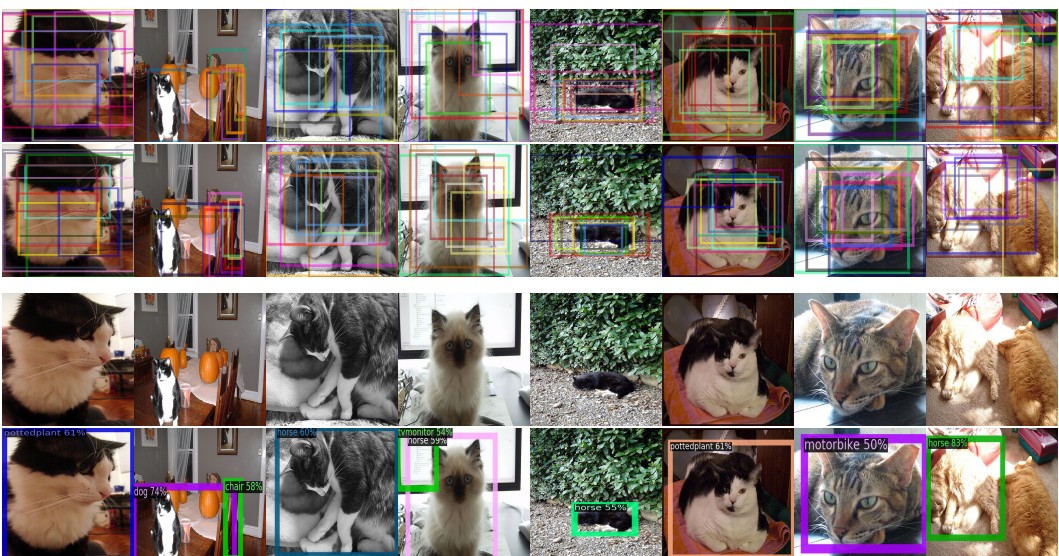

Figure 5: Top 10 proposals and final detection results of TFA (Wang et al., 2020) (proposal **row 1**; detection **row 3**) and CoRPNs (proposal **row 2**; detection **row 4**). All images here are **cat** (cat is a novel category) test images. Due to the proposal neglect effect, state-of-the-art TFA misses all cat boxes. By contrast, CoRPNs catch many more cat objects by eliminating the effect, thus significantly outperforming TFA. However, we also notice that CoRPNs still have difficulty of correctly classifying the proposals as cats, suggesting that improving the classifier will further boost the detection performance, which we leave as future work.

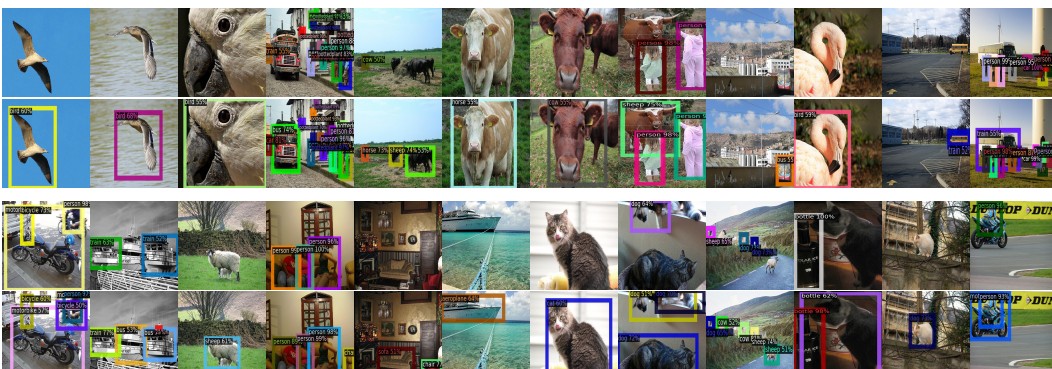

Figure 6: Detection results of TFA (Wang et al., 2020) (**row 1 & 3**) and CoRPNs (**row 2 & 4**) under PASCAL VOC split 1 shot 1 (**row 1 & 2**), and split 3 shot 1 (**row 3 & 4**). Novel classes shown here are {bird, bus, cow, boat, cat, motorbike, sheep, sofa}. TFA fails to detect many novel objects due to the proposal neglect effect, while our CoRPNs catch many more novel objects by eliminating the effect.

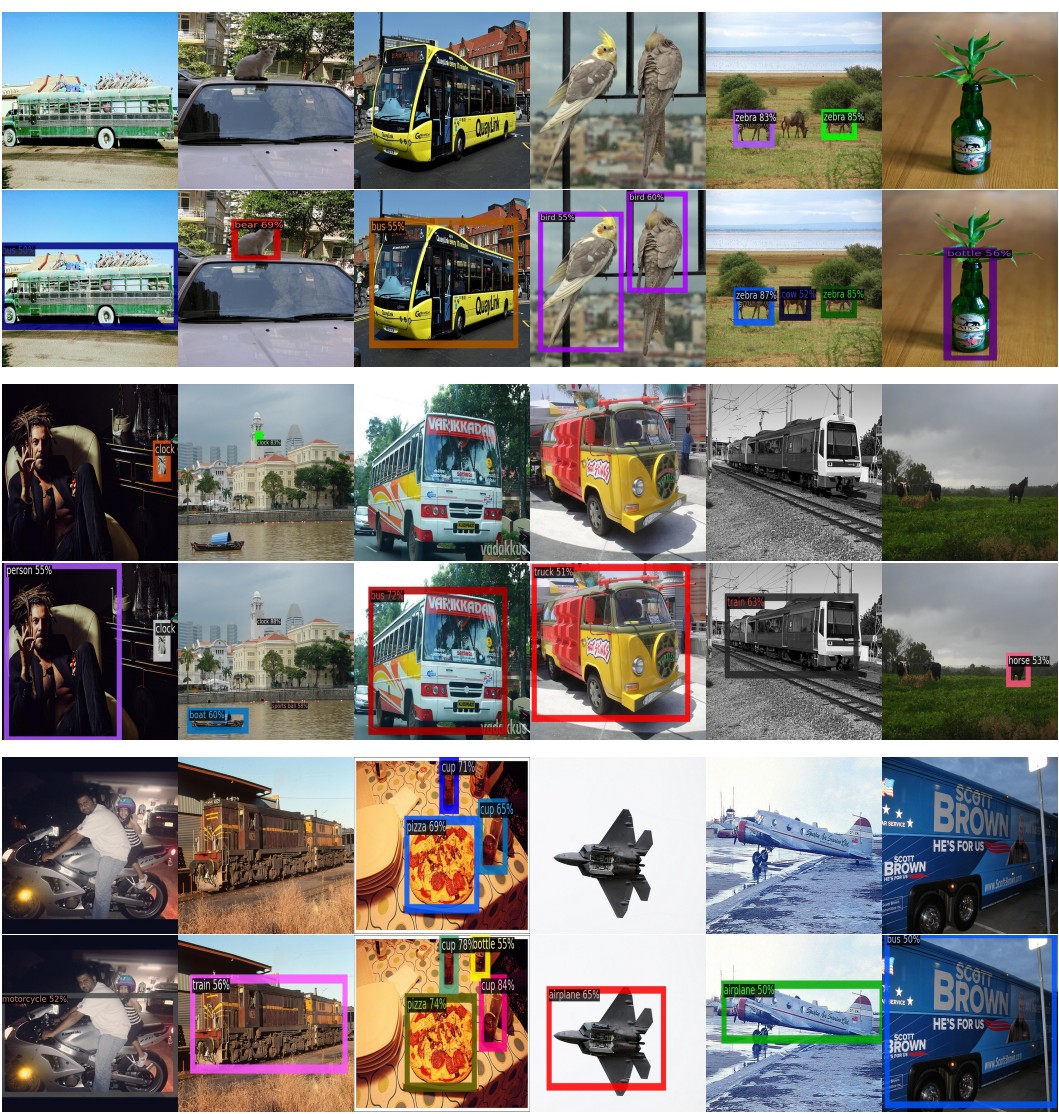

Figure 7: Detection results of TFA (Wang et al., 2020) (**row 1 & 3 & 5**) and CoRPNs (**row 2 & 4 & 6**) under COCO test sets (val2014). Novel classes shown here are {airplane, bird, boat, bottle, bus, cat, cow, horse, motorcycle, person, train}. TFA fails to detect many novel objects due to the proposal neglect effect, while our CoRPNs catch many more novel objects by eliminating the effect.

# F PASCAL VOC VISUALIZATIONS

We provide novel object detection result visualizations of CoRPNs and TFA (Wang et al., 2020) on PASCAL VOC in Figure 6. TFA reflects the proposal neglect effect and misses lots of novel objects, while CoRPNs detect many more novel objects.

# G COCO VISUALIZATIONS

We provide additional novel object detection result visualizations of CoRPNs and TFA (Wang et al., 2020) on COCO in Figure 7. Again, TFA fails to detect many novel objects due to the proposal neglect effect, whereas CoRPNs catches many more novel objects by eliminating the effect.

