# OpenReview forum: "Cooperating RPN's Improve Few-Shot Object Detection"
_ICLR.cc/2021/Conference — Reject_

### Official Review · AnonReviewer3 · 2020-10-16
**Interesting idea, clearly-written paper, but there are some analyses missing (review updated)**

**Rating:** 5
**Confidence:** 4

**Review:**

In this paper, the authors present a interesting, novel idea of promoting the diversity and cooperation among multiple RPNs for the problem of few-shot detection. They first identify a critical problem in few-shot detection, which is that existing RPNs can miss objects of novel classes and that their proposed boxes make very similar errors. In order to resolve this issue, the authors propose to utilize multiple RPNs in the same detection pipeline and include a diversity loss when training the RPNs, such that they provide diverse scores to the anchor boxes. In this way, the chance of all the RPNs missing an object becomes small. The authors further include a cooperation score to enforce the RPNs to provide more meaningful scores (rather than sparse scores just to promote diversity). The proposed approach is evaluated on COCO and Pascal, and the results demonstrate that the proposed approach does improve the detection performance, as compared to the state of the art, especially in very low-shot cases.

Overall, this is a very well-written paper. The authors have identified the existing problem and motivated their work well. The proposed method is novel and well described, and the evaluation is thorough.

There is one thing that I am curious about and would hope to get the authors' insights. In the evaluation, the authors compare their proposed method with a naive ensemble approach, which combines the results of RPNs with different random initializations.  I am curious about the performance of a slightly less naive, yet standard ensemble method, i.e., bootstrapping the training set. This could potentially allow the RPNs to learn from slightly different data and then provide diverse outputs as well.

I don't have any other major comments. Here are a few minor ones.

- For the diversity loss, did the authors try only penalizing the case when the determinant is close to zero, as compared to making it large? It seems that the sign of the determinant shouldn't matter. The current choice is reasonable and results in a good detector, but I am just curious what the authors think.

- How does one determine a good number of RPNs in general? The authors mention that this can be problem/dataset-dependent. In practice, should one rely on a validation set to choose this number? Or is there any exploratory analysis that one could perform to find a good (mostly not optimal) number?

- Please increase the gap between Tables 4 and 5.


----------- updates after reading author response -----------

After looking at the authors' response, the revised paper, and the other reviews, I'd like to update my rating from "Accept" to "Marginally below acceptance threshold". While I still like the overall idea of this paper and I believe this is an interesting direction of research, the additional results in the revised paper actually raise more questions and there are issues that need to be addressed in this current study.

1. Properly quantifying "proposal neglect"

The improvements shown in Table 3 are very small, which do not support the claim that the proposed method produce "more and better boxes". Table 3 might actually not be the correct way to assess the "proposal neglect" (or resolving proposal neglect). I would expect to see a more in-depth result analysis. For instance, when comparing with the SOTA TFA approach, for each image on average, how many more correct objects are detected due to the proposed approach being able to find boxes that TFA cannot. Something like this would clearly indicate the benefit of the proposed approach.

2. Sensitivity to hyperparameters

In Fig. 4 (left), there is a huge sensitivity to the hyperparameter selection. For instance, the performance of the proposed approach would drop drastically by just changing the number of RPNs from 2 to 3. This creates uncertainty for people to use this approach. Furthermore, for the COCO results, when the hyperparameters are not selected based on this set, the improvements are pretty small as compared to TFA. I believe a more robust hyperparameter selection method would be needed for the proposed approach to thrive.

---

> ### Author Response · Authors · 2020-11-25
> **response to AnonReviewer3**
>
> Thank you for reviewing our paper and providing helpful feedback.  We address all the points as follows.
>
> 1. “bootstrapping baseline”:
>
> Thanks for the suggestion. We add an ablation in the revised submission (**Table 6**) to compare with a bootstrapping baseline. We construct the bootstrapping baseline also using multiple RPN classifiers like in CoRPNs. During training, instead of selecting the most certain RPN, we randomly select an RPN to get gradients. During the fine-tuning and inference time, we use the same selection procedure as CoRPNs (for each box, use the most certain RPN). CoRPNs outperform the bootstrapping baseline, suggesting that *RPN variance obtained by training data variation does not lead to much improvement*.
>
> 2. “determinant loss”:
>
> Thanks for the observation. Since the covariance matrix is positive semi-definite, the determinant is always non-negative. By the Gershgorin circle theorem, the determinant is bounded since the largest eigenvalue is bounded. In this case, making a large determinant forces the PCA variance to spread over all RPN’s. If we could have a global matrix containing RPN's responses to *all* foreground boxes, we want this matrix's determinant to be as large as possible. But since the log determinant loss is calculated *at each training batch, not at all training data*, we do not need to make the determinant large for every single batch.
>
> 3. “Determine a good number RPNs”:
>
> First, as shown in updated Figure 4 of the revised submission (the **middle** subfigure), using selected hyperparameters, the number of RPN's does not affect the result much. CoRPNs with any number of RPN's outperform TFA (number of RPN = 1). Second, as explained in detail in the two newly added paragraphs (with title **Hyperparameters on PASCAL VOC** on page 5 and **Hyperparameters on COCO** on page 6) of the revised paper, we selected hyperparameters (including the number of RPN’s) using some criteria *on base classes*. In addition,  *We did not select hyperparameters for the challenging COCO dataset. Instead, for COCO, we selected hyperparameters that worked well on all three splits in PASCAL VOC and directly used those*. We can use a similar procedure in practice.
>
> 4. “Increase the gap between Tables 4 and 5”:
>
> Thanks for the comment. We updated the table layout in the revised paper.
>
> Gershgorin circle theorem: https://en.wikipedia.org/wiki/Gershgorin_circle_theorem

---

### Official Review · AnonReviewer2 · 2020-10-28
**Few shot object detection using multiple region proposal networks that specialize in their own domains, reaching some SotA results**

**Rating:** 7
**Confidence:** 3

**Review:**

The authors propose a few-show object detection architecture, which improves the 1st stage of two-stage detectors (R-CNN in particular). In a few shot setting, the existing approaches the region proposal generator may ignore some novel out-of-distribution classes as they have not been included at training time. The proposed method attempts to correct this by using many RPN's, and training them such that the gradient is only passed to one of them at a time; thus forcing them to learn mutually different kinds of regions.

The topic is relevant, and not too widely studied. Few-shot learning is well presented in the literature in the classification setup, but less so in detection. The idea is novel, although quite straightforward and somewhat arbitrary. I encourage the authors to clarify the insight as to why this particular combination of losses was chosen (I could think of alternative formulations). Also it would be important to see some discussion about the computational cost: for object detection, people often prefer simple and fast approachs (SSD, Yolo) that may be less accurate but are fast to execute. How much does the proposed method with N classifiers increase computation (something less than N-fold, but how much)?

The results reach state-of-the-art on some cases; especially with very-few-shot domain.

---

> ### Author Response · Authors · 2020-11-25
> **response to AnonReviewer2**
>
> Thank you for reviewing our paper and providing helpful feedback.  We address all the points as follows.
> 1. “why this particular combination of losses was chosen”:
>
> First, we do not claim that our loss is the only feasible way of training diverse but cooperating RPN’s. We construct our loss in this way to achieve two goals: (1) To ensure that RPN’s are different; there are indeed many ways of making so, depending on how to determine if an RPN is different from another RPN. In our scenario, “different” means that *each RPN makes a different set of predictions for foreground boxes*. More concretely, we want the $M$*$N$ ($M$ number of foreground boxes, $N$ number of RPN's) response matrix to have rank $N$. The way we construct our divergence loss is a straightforward way to achieve this. (2) To ensure that no RPN gives an extremely low score on a foreground box. Given our RPN selection procedure, the score of a box is determined by the most certain RPN. If an RPN produces a close to zero score to a foreground box (which will happen given only the divergence loss), then it is likely that CoRPNs misclassified the foreground box as background. A straightforward way of avoiding the above is to add a margin loss to ensure that every RPN at least produces a score of $\phi$ on every foreground box, which is our cooperation loss.
>
> 2. “Computational cost”:
>
> Thanks for the comment. We extend a section in the Appendix (**Training Time and Memory** on page 12) to discuss the training time and memory cost of CoRPNs. First, CoRPNs are constructed by “redundant classifiers while keeping both the feature extractor and the bounding box regressor shared between all RPN's” (as mentioned in the Section “Learning Cooperating RPN’s” and illustrated in Figure 3). Therefore, the only architectural difference between CoRPNs and TFA is that CoRPNs (with $N$ classifiers) have ($N-1$) more RPN classifiers. In TFA, the RPN classifier is a 1*1 convolutional layer with input channel 256 and output channel as the number of anchors per position (3 in TFA). For CoRPNs with $N$ RPN classifiers, the number of added parameters is 256 * 3 * ($N-1$). With this, we find that CoRPNs with 5 RPN's increase the training time by 3\%, with roughly the same memory compared with TFA.

---

### Official Review · AnonReviewer1 · 2020-10-28
**Using multiple RPNs to improve few-shot object detection**

**Rating:** 6
**Confidence:** 5

**Review:**

Summary:
The paper argues that in the case of few shot object detection, the quality of region proposals is important, as if the region proposal network misses any of the very scarce positive box, the performance is severely impacted. It proposes to use multiple cooperating RPNs to alleviate the problem and improve few shot detection.


Positives
- The paper is easy to follow and proposes a simple idea of improving the RPN classifiers so that any proposal with high IOU with the object is not pruned out early due to the RPN. At the same time the RPN maintains pruning performance and does not send too many false positives.
- The method proposed is clear
- The evaluations are done on standard challenging benchmarks for the problem and good performances are reported

Negatives
- The thesis is simple and acceptable, i.e. for few shot detection training, missing any high IOU proposal can be very bad for performance, so improve the RPN to avoid that happening at that stage, however the proposed method is the best way to do it is not convincing. Why does what is being proposed to solve this problem of proposal neglect the right way to do so?
- In particular there can be many other ways of making the RPN classifier better, e.g. by simply making the sub network bigger, which would be fair as in the proposed method also more parameters are added
- There seems to be no empirical proof with ablation experiments, that the diversity and cooperation losses are useful for that task? What happens when we train the same without using, either or both of the losses (with acceptable settings of parameters)
- In fig 4 all results with number of RPNs from 1 to 10 should be reported. The sudden jump from 5 to 10 seems a bit convenient for the claim that large number of RPNs lead to large number of FPs leaking reducing the performance. The performances of naive RPN ensembles can also be added to the bar plot (instead of selected numbers in tab. 4)
- The choice of number of RPNs should be properly cross-validated/justified. Right now it feels a bit ad-hoc; in supplementary it is mentioned "We mostly use five RPN’s, except for PASCAL VOC novel split 3, where two RPN’s lead to better performance." -- has this been tuned based on the performance on the test sets?
- The hypothesis that large number of FPs degrade the performance for large number of RPNs, can be evaluated empirically also by computing the actual statistics for region proposals accepted or rejected.
- Own baseline (same implementation details/strategy as CoRPN) with #RPN=1 should be reported for all tables. Right now the point of reference seems to be TFA, which is implemented elsewhere; the implementation here might already have slight benefits due to the parameter settings etc.

Overall, the thesis is acceptable but the method is still not very convincing. The performances reported are state of the art in COCO (more for very few shots 1,2 etc. and marginal for larger), marginal for VOC low shots and not for higher shots. I am a bit lukewarm about the paper in general.

---

> ### Author Response · Authors · 2020-11-25
> **response to AnonReviewer1**
>
> Thank you for your efforts in reviewing our paper and providing constructive feedback.  The comments focus mostly on the clarity of some details and additional baseline comparisons. We address all these points as follows.
>
> 1. “Why is CoRPNs the right way to do solve proposal neglect?”:
>
> We thank the reviewer for the comment. As mentioned in the Conclusion Section of the submission, “we don't claim that our method is the best way to control proposal neglect.” Here we would like to clarify the contributions of this paper:
> (1) identify an important yet under-explored issue (i.e., proposal neglect) in few-shot detection;
> (2) provide one straightforward yet effective strategy to address proposal neglect (i.e., CoRPNs), as demonstrated by experiments and analysis.
> As shown in Table 3, small improvements in proposal neglect yield significant performance improvement; therefore, further investigation of better strategies is an excellent direction for future research.
>
> 2. “Compare with a bigger RPN sub-network”:
>
> Thanks for the comment. We add an ablation in the revised submission (**Table 7**) to compare CoRPNs with baselines using larger RPN sub-networks. We find that using larger RPN sub-networks does not improve performance, suggesting that the advantage of CoRPNs is *not simply the result of using more parameters*. We would also like to clarify that CoRPNs “construct redundant classifiers while keeping both the feature extractor and the bounding box regressor shared between all RPN's” (as mentioned in the Section “Learning Cooperating RPN’s” and illustrated in Figure 3). Therefore, CoRPNs do not significantly increase the number of parameters. As discussed in the extended **Training Time and Memory** section (Appendix, page 12), CoRPNs occupy roughly the same memory compared with TFA.
>
> 3. “using only the diversity loss or only the cooperation loss”:
>
> Thanks for the comment. We add an ablation in the revised submission (**Table 5**) to discuss results using one loss term. We find that using either loss alone does not improve the performance, indicating our diversity loss and cooperation loss are *both* required for CoRPNs to obtain improvements.
>
> 4. “Figure 4 with all numbers from 1 to 10, as well as naive RPN ensembles”:
>
> Thanks for the comment. As suggested, we update Figure 4 in the revised paper. The **right** subfigure in updated Figure 4 shows the results of naive RPN ensembles with different numbers of RPNs, ranging from 1 to 10. We find that naive RPN ensembles improve the performance but not as good as CoRPNs with selected hyperparameters (the **middle** subfigure).
>
> 5. “how do we choose the number of RPN’s”:
>
> First, as shown in updated Figure 4 in the revised submission (the **middle** subfigure), using selected hyperparameters, the number of RPN's does not affect the result much. CoRPNs with any number of RPN's outperform TFA (number of RPN = 1). Second, as explained in detail in the two newly added paragraphs (with title **Hyperparameters on PASCAL VOC** on page 5 and **Hyperparameters on COCO** on page 6) of the revised submission, we selected hyperparameters (including the number of RPN’s) using some criteria *on base classes*. For VOC split 3, we found that using 2 RPN’s already led to a high ‘Avg # FG’ in the base classes, so we used 2 RPN's. (Here, ‘Avg # FG’ is the number of foreground boxes generated by the RPN, which we used as a metric to assess the impact of proposal neglect.)
>
> 6. “The hypothesis that large number of FPs degrade the performance for large number of RPNs, can be evaluated empirically also by computing the actual statistics for region proposals accepted or rejected”
>
> With updated Figure 4 in the revised submission, we would like to revise our statement about the number of RPN's accordingly. We find that the performance degradation in using a larger number of RPN’s was because of *using the same set of hyperparameters ($\phi$, $\lambda_{d}$, and $\lambda_{c}$)*. When using selected hyperparameters for different numbers of RPN's, we can also achieve good results with a larger number of RPN’s (like 10).
>
> 7. “Baseline with #RPN=1 should be reported”:
>
> #RPN=1 is TFA. We used the publicly released code by TFA authors to implement CoRPNs. As mentioned in the Experiments Section, “we used the same values of all shared training and fine-tuning hyperparameters (batch size, learning rate, momentum, weight decay, etc.) as TFA. Therefore, CoRPNs *did not obtain any benefit from implementation and parameter settings*. We will also release our code upon acceptance.

---

### Official Review · AnonReviewer5 · 2020-11-04

**Rating:** 3
**Confidence:** 4

**Review:**

Post-review comments:

After reading the reply I decided I will keep my low score though it hurts to do so for a paper into which the authors definitely invested a lot of energy. Here is the reason why:

I think there are usually two ways in which a paper can make an important contribution: Through a new insight or through a new piece of modelling that will be widely used afterwards. I think in it's current form the paper presents neither.

The potential insight I see is proposal neglect. However even with the added experiments in Table 3 I find the results neither sufficient to prove the effect exists. The GT boxes are already added so overlooking objects while fine-tuning should not be a big issue. If it was it should have a bigger impact (improvements in Table 3 and on general performance are minimal).

So what about a widely usable piece of modelling? Despite the anecdotal motivation the presented method improves performance. So the question becomes: Will this be widely used? My prediction is that it won't. The improvement is rather incremental but the effort to use it is high. While the model is simple it comes with three additional hyperparameters which the authors tune individually for each experiment (Section 4, Hyperarameters VOC & COCO). This is the biggest issue I have with the method: It requires tuning a bunch of hyperparameters to achieve a marginal improvement.

Compare this to TFA [1], the method the paper builds upon. TFA is built upon a very simple insight (fine-tuning the heads works better than comparing representations) and because the insight and model are easy to use they will be the basis for a number of follow-up works (e.g. this paper). I cannot see the same happen with the presented method as long as hyperparameters have to be tuned for each dataset and split. I am sorry but in my eyes this is bad practice!.

To me this means the method will likely be of no lasting value. While it is SotA in the one-shot case for now I think the insights and methods used for achieving this performance cannot be used by other groups to improve performance even more.

So what can I recommend the authors to do with the paper? I think there are two ways to increase the papers contribution:
1. Study different effects and problems of RPNs in few-shot object detection. A better understanding will for sure help moving the problem forward. Even if the end result is: The RPN works surprisingly well. That would be a great insight as well in my eyes as it would free resources to address the other problems.
2. Improve the method so it provides a significant gain without any additional hyperparameter tuning. At only 3% more computing time using it would probably be a no-brainer if it was not for the hyperparameter tuning.

[1] Wanget. al ICML 2020, "Frustratingly simple few-shot object detection"


-------------------------------
1. Summary

The paper addresses the region proposal network (RPN) as a key component in few-shot object detection. Their analysis is centered around the concept of "proposal neglect": objects in the meta training set are ignored because no proposals are generated for them. They propose CoRPN, an assemble of RPN classifiers, to tackle the problem and present results on the original few-shot COCO and Pascal VOC benchmarks as well as ablation results.


2. Strengths
+ I really like the general motivation and the RPN is for sure a key component for good few-shot object detection.
+ The proposed model and especially the diversity loss seem well chosen to address "proposal neglect".
+ The method brings a small but consistent performance improvement over TFA.

3. Weaknesses
- I think the main flaw of the paper is that there is no quantitative analysis of "proposal neglect", only some qualitative examples in the appendix. I don't want to say that it does not exist it makes only sense that such an effect messes up meta training. But I am missing experiments demonstrating the effect and it's impact on performance.
- If proposal neglect is such a big issue for meta-training, why not simply add the ground truth boxes as proposals? Many object detection frameworks have that option built in. Comparison to this baseline seems crucial to evaluate the value of the proposed method.
- The new hyperparameters (N-rpn, phi, lambda_c, lambda_d) are apparently not fixed but chosen in a per-experiment basis as only ranges are provided in the appendix, not exact values. This makes it hard to judge the actual value of the method as hyperparameters have to be chosen in advance in real-world applications.
- I find the ablations lacking for a number of reasons which is critical as this should be the section that allows assessing the impact of "proposal neglect" and the proposed method. Specifically I have three issues:
-- I am not sure the proposed experiments are suited to grasp the effect of the original problem and the effect of the method. How does the method impact RPN recall? What about classification accuracy, false negatives etc.? Table 6 does attempt to provide this information but I find it hard to understand what exactly is reported. Probably what I am interested in is already there it's just not that clear yet?
-- Ablations have only been performed on Pascal VOC and not on the more interesting and more challenging COCO dataset. Especially Table 6 would be interesting for COCO as well.
-- Ablation results appear to be extremely noisy. Compare the differences between configurations in Figure 4, adding or removing a classifier should not lead to a 5% performance change especially if this effect is not consistent across splits and changes again when another classifier is added or removed. To be clear this is not the authors fault but a problem of the setup introduced in Kang et. al. 2018 using a fixed set of images for meta-training. This approach makes overfitting to the specific chosen examples a big concern. Now I think it is good the authors follow the established evaluation protocol but this design makes it rally hard to examine if results are valid. In the case of the presented ablation studies this effect definitely casts a large shadow of doubt on the significance of these ablations.

4. Recommendation

I think in it's current form this paper has to be rejected. At first I was excited to see someone addressing proposals in few-shot object detection but the paper sadly only addresses "proposal neglect" without any quantitative analysis or consideration of other potential problems. Furthermore due to the potentially large variability of results due to the fixed meta-training set (see variability in Figure 4) and potential hyperparameter variations between experiments it is hard to judge if the reported performance gains are robust and could be transferred to real-world applications. I do feel an investigation of the RPNs role in few-shot object detection is very worthwhile and the proposed method may in fact be valuable, however to make either of these two contributions a deeper analysis is needed.

5. Questions/Suggestions
- Provide quantitative analysis of the "proposal neglect" problem. Even better: Do a throughout assessment of the RPNs role and problems in few-shot object detection.
- Compare to a simple baseline: Adding the gt bounding boxes to the proposals during meta-training
- Please clarify if hyperparameters were the same for all experiments and report the chosen values in the main paper, not the appendix.
- Run ablation experiments on COCO as well
- Better explain Table 6

6. Additional feedback

None, the paper is overall well written, figures are clear and understandable.

---

> ### Author Response · Authors · 2020-11-25
> **response to AnonReviewer5, part 1**
>
> Thank you for your efforts in reviewing our paper and providing insightful feedback.
> The comments focus mostly on quantitative analysis of the proposed “proposal neglect” effect and impact of hyperparameter variations. Below, we first address these most pressing concerns and then address all the other points.
> **Main concern 1 -- “quantitative analysis of the ‘proposal neglect’ problem”**
> We thank the reviewer for the comment on using *quantitative* analysis to assess the impact of “proposal neglect”. In fact, as mentioned by the reviewer, we already used “the number of foreground boxes generated by the RPN after non-maximum suppression (Avg # FG)” as a quantitative metric to analyze the “proposal neglect” effect and its impact on the final detection performance, though such an analysis was mainly conducted to investigate the threshold hyperparameter $\phi$ in our proposed cooperation loss (Table 6 in the original submission). In the revised submission, we *extend the use of this quantitative metric to systematically* understand the “proposal neglect” effect and compare the TFA baseline and our CoRPNs.
>
> Specifically, we add a new paragraph (with the title **Is Proposal Neglect Occurring?** on page 6) and **Table 3** to provide a quantitative analysis of the proposal neglect effect. As mentioned earlier, ‘Avg # FG’ is a direct metric reflecting the proposal neglect effect. ‘Avg # FG’ can also be viewed as the number of boxes *"recalled"* from the RPN. When an RPN suffers from the proposal neglect effect, ‘Avg # FG’ is lower and the RPN generates fewer foreground boxes. At the fine-tuning stage (fine-tuning with novel classes), an RPN suffering from the proposal neglect effect outputs fewer novel class boxes; therefore, the corresponding classifier cannot build an accurate model of appearance variation for novel classes, leading to degraded detection performance.
>
> In Table 3, we present 'Avg # FG' at three IoU thresholds produced by CoRPNs and TFA at the novel class fine-tuning stage. Table 3 shows that *CoRPNs produce more and better foreground boxes than TFA*. We expect these small improvements in the box proposals to have a large effect, because such boxes improve the classifier’s model of appearance variation for novel classes; small changes in this model can *result in significant movements of the class boundaries.*

---

> > ### Author Response · Authors · 2020-11-25
> > **response to AnonReviewer5, part 2**
> >
> > **Main concern 2 -- “Due to … potential hyperparameter variations between experiments, it is hard to judge if the reported performance gains are robust and could be transferred to real-world applications”**
> >
> > First, we would like to clarify the selection procedure of the hyperparameters and their values. Here are the hyperparameter values used in all reported results:
> >
> > VOC split 1: 5 RPNs, $\phi$ = 0.2, $\lambda_{c}$ = 2, $\lambda_{d}$ = 0.025.
> > VOC split 2: 5 RPNs, $\phi$ = 0.1, $\lambda_{c}$ = 1, $\lambda_{d}$ = 0.025.
> > VOC split 3: 2 RPNs, $\phi$ = 0.5, $\lambda_{c}$ = 1, $\lambda_{d}$ = 0.3.
> > COCO: 5 RPNs, $\phi$ = 0.3, $\lambda_{c}$ = 1, $\lambda_{d}$ = 0.025.
> > Note that
> > (1) The hyperparameter values were *not chosen on a per-experiment basis*. Instead, for the same dataset, we used a *single* set of hyperparameters, regardless of the number of shots in novel classes. (For VOC, different splits correspond to different sets of base and novel classes, so we used different hyperparameter values).
> > (2) *We did not select hyperparameters for COCO. Instead, we selected hyperparameters that worked well on all three splits in VOC and used those hyperparameters on COCO*. This shows the robustness and generalizability of our approach.
> >
> > As suggested, we add two new paragraphs in the revised paper (with title **Hyperparameters on PASCAL VOC** on page 5 and **Hyperparameters on COCO** on page 6), which report the selected values. Besides, the reason why “in the original appendix, we provided ranges of hyperparameter values instead of specific numbers” was because we found that the performance within the given ranges is stable.
> >
> > In these two new paragraphs, we also explain in detail how we selected hyperparameters. As a summary, for PASCAL VOC, we selected hyperparameters *using base classes* with the following criterias:
> > (1) The average number of RPNs responding positively to a foreground box (a larger average means that the RPN's are cooperating well with each other by responding positively to foreground boxes).
> > (2) The cumulative variance in the RPN's response matrix to foreground boxes (please find details on how to do this in the revised Appendix section *Selection Procedure – Cumulative Variance*. If the cumulative variance is large in the first few components, RPN's are responding similarly to foreground boxes).
> > (3) The average number of positive samples (larger means that the CoRPNs produce more high IOU boxes).  *Because the hyperparameters were selected on base classes, we can use a different set per split without concerns about generalization.*
> >
> > Second, we would like to clarify that the performance improvement of our CoRPNs is not caused by hyperparameter variance. As mentioned before, for the challenging COCO dataset, we selected hyperparameters on PASCAL VOC and used those. This means that improvements in COCO are not caused by hyperparameter variance. Moreover, we report results for another two sets of hyperparameter (also selected from PASCAL VOC) in **Table 9** (in the revised Appendix). We also report the mean, variance, and 95% confidence intervals obtained by the three hyperparameter sets. Table 9 shows that CoRPNs outperform TFA with different sets of hyperparameters and the improvements are significant.

---

> > > ### Author Response · Authors · 2020-11-25
> > > **response to AnonReviewer5, part 3**
> > >
> > > **Response to other comments:**
> > >
> > > 1. “Compare to a simple baseline: Adding the gt bounding boxes to the proposals during meta-training”:
> > >
> > > Following TFA, we already appended all the ground-truth boxes to the proposals as training examples in all the experiments. We did this for both TFA and CoRPNs. We realized that this was confusing, and in the revised submission, we add two sentences at the end of the **Datasets and Implementation Details** to clarify.
> > > As reported in the TFA paper, adding the ground-truth boxes brings a 0.5% AP gain on COCO. However, with limited ground-truth boxes for novel classes in the few-shot regime, they are *not enough* for the classifier to build a good model of appearance variation for novel classes. As discussed in the response to main concern 1, CoRPNs produce more high IOU boxes other than the ground-truth boxes, which helps the classifier to build a better model of variation for novel classes.
> > >
> > > 2. “Better explain Table 6”:
> > >
> > > Thanks for the comment. We realized that the caption of Table 6 was confusing. For clarity, we update it, and now Table 6 becomes **Table 8** in the revised submission. In Table 8, ‘Avg # FN’ is the average number of foreground training boxes misclassified by the RPN (calculated before non-maximum suppression). ‘Avg # FG’ is the average number of foreground boxes produced by the RPN (after non-maximum suppression). The goal of Table 8 is to investigate $\phi$'s impact on training RPN's to cooperate, and for this purpose, we mainly focus on the metric of ‘Avg # FN.’ In the cooperation loss, $\phi$ is the lower bound of each RPN's response to foreground boxes. When $\phi$ increases, we expect ‘Avg # FN’ to decrease since CoRPNs become more and more unlikely in misclassifying foreground boxes (CoRPNs get a foreground box wrong only when the most certain RPN thinks the box is background). Our expectation is consistent with the results in Table 8: as the value of $\phi$ increases, ‘Avg # FN’ indeed decreases. In addition, lower Avg # FN’ leads to higher ‘Avg # FG’ after NMS, and CoRPNs with all $\phi$ values also have a larger ‘Avg # FG’ compared to the TFA baseline.
> > >
> > > 3. “Variance in Figure 4”:
> > >
> > > Thanks for the comments. We realized that Figure 4 was confusing. For clarity, we update Figure 4 to better analyze the effect of different numbers of RPN's. In the original Figure 4 (which is now the *left* subfigure in the updated version), we showed the results of *using the same set of hyperparameters* ($\phi$, $\lambda_{d}$, and $\lambda_{c}$). We find that there is considerable variance across the number of RPN's. In the *middle* subfigure of the updated version, we show the results of *using selected hyperparameters* (with selection procedures discussed in response to main concern 2) for different numbers of RPN's. CoRPNs with any number of RPN's outperform TFA (number of RPN = 1), and the variance is much smaller than on the left subfigure (i.e., the original Figure 4). Therefore, with selected hyperparameters, the number of RPN's does not affect the result much.
> > >
> > > 4. “Ablation experiments on COCO”:
> > >
> > > In the newly added Table 3 of the revised submission, we provide analysis on COCO to quantitatively investigate the proposal neglect effect and its impact on few-shot detection performance, as well as comparing TFA and CoRPNs. In addition, in Table 9 in the revised Appendix, we also provided ablation showing that CoRPNs improves TFA in COCO with different hyperparameters.
> > >
> > > While it is interesting to have more ablations on COCO, we believe that our current ablations (most on PASCAL and some on COCO) are sufficient to show the effectiveness and impact of different components and design choices of CoRPNs. In particular, as mentioned in the response to main concern 2, we selected hyperparameters that worked well on VOC and directly used those on COCO. In addition, conducting more ablations on COCO was constrained by time and our limited GPU computation resources. For example, running Figure 4 and Tables 5, 6, 7, 8 on COCO all need to run experiments on both stage 1 (base classes training) and stage 2 (novel classes fine-tuning), which are time-consuming and GPU expensive.

---

### Public Comment · ~Alaa_M_El-Nouby1 · 2020-11-12
**Finetuning RPN baseline**

Thanks for the authors for their interesting work. I had a question regarding the baselines, I am wondering if you have tried a baseline where a single RPN is simply finetuned, similar to the classifier and the box regression heads in TFA ?

---

> ### Author Response · Authors · 2020-11-25
> **fine-tuning RPN baseline**
>
> Thank you for your comment and interest in our paper. We ran experiments on fine-tuning RPN for all three PASCAL novel splits under the 1-shot setting. We used the publicly released code by TFA authors (https://github.com/ucbdrive/few-shot-object-detection), where they provide the option of fine-tuning RPN. We also need to fine-tune the box head (the feature extractor in the ROI head) after the RPN. We found that fine-tuning the RPN degrades performance in split 1 & 2 and improves performance in split 3. Our CoRPNs substantially outperform this RPN fine-tuning baseline in all splits. The detailed results are:
> PASCAL VOC, novel classes AP50, 1-shot, cosine classifier
>
> $\hspace{4cm} $Novel Set 1$\hspace{2cm} $Novel Set 2$\hspace{2cm}$Novel Set 3
> TFA$\hspace{4cm}$39.8$\hspace{3cm}$23.5$\hspace{3cm}$30.8
> TFA w/ fine-tuned RPN$\hspace{1.2cm}$36.9$\hspace{3cm}$20.7$\hspace{3cm}$34.1
> CoRPNs$\hspace{3.4cm}$**44.4**$\hspace{2.9cm}$**25.7**$\hspace{2.9cm}$**35.8**

---

### Author Response · Authors · 2020-11-25
**Thanks for all your comments & updated the paper to a revised version**

We thank all the reviewers for their insightful comments and suggestions. We address all the comments below and have also updated the submission accordingly, with the following major revisions:

(1) New section **“Experiments -- Is Proposal Neglect Occurring?”** provides quantitative analysis on the proposal neglect effect.
(2) New sections **“Experiments -- Hyperparameters on PASCAL VOC”** and **“Experiments -- Hyperparameters on COCO”** discuss how we selected hyperparameters.
(3) We added several new ablations. For example, Table 5 investigates using only diversity loss or cooperation loss; Table 6 compares with a bootstrapping baseline; Table 7 compares with baselines with larger RPN sub-networks; Updated Figure 4 presents naive ensembles with number of RPNs from 1 to 10 and CoRPNs with selected hyperparameters; Updated caption in Table 8.

---

### Decision · Program_Chairs · 2021-01-07
**Final Decision**

**Decision:**

Reject

**Comment:**

The reviewers have not supported the acceptance of this paper where the key weakness is that the study of the proposal neglect effect is not sufficient (see the reviews for the details). I agree with the assessment of the reviewers and recommend rejecting the paper in its current form.